# A no-regret generalization of hierarchical softmax to extreme multi-label classification

**Marek Wydmuch**
Institute of Computing Science
Poznan University of Technology, Poland
mwydmuch@cs.put.poznan.pl

**Kalina Jasinska**
Institute of Computing Science
Poznan University of Technology, Poland
kjasinska@cs.put.poznan.pl

**Mikhail Kuznetsov**
Yahoo! Research
New York, USA
kuznetsov@oath.com

**Róbert Busa-Fekete**
Yahoo! Research
New York, USA
busafekete@oath.com

**Krzysztof Dembczyński**
Institute of Computing Science
Poznan University of Technology, Poland
kdembczynski@cs.put.poznan.pl

## Abstract

Extreme multi-label classification (XMLC) is a problem of tagging an instance with a small subset of relevant labels chosen from an extremely large pool of possible labels. Large label spaces can be efficiently handled by organizing labels as a tree, like in the hierarchical softmax (HSM) approach commonly used for multi-class problems. In this paper, we investigate probabilistic label trees (PLTs) that have been recently devised for tackling XMLC problems. We show that PLTs are a *no-regret* multi-label generalization of HSM when precision@$k$ is used as a model evaluation metric. Critically, we prove that *pick-one-label* heuristic—a reduction technique from multi-label to multi-class that is routinely used along with HSM—is not consistent in general. We also show that our implementation of PLTs, referred to as EXTREMETEXT (XT), obtains significantly better results than HSM with the pick-one-label heuristic and XML-CNN, a deep network specifically designed for XMLC problems. Moreover, XT is competitive to many state-of-the-art approaches in terms of statistical performance, model size and prediction time which makes it amenable to deploy in an online system.

## 1 Introduction

In several machine learning applications, the label space can be enormous, containing even millions of different classes. Learning problems of this scale are often referred to as *extreme classification*. To name a few examples of such problems, consider image and video annotation for multimedia search (Deng et al., 2011), tagging of text documents for categorization of Wikipedia articles (Dekel & Shamir, 2010), recommendation of bid words for online ads (Prabhu & Varma, 2014), or prediction of the next word in a sentence (Mikolov et al., 2013).

To tackle extreme classification problems in an efficient way, one can organize the labels into a tree. A prominent example of such label tree model is *hierarchical softmax* (HSM) (Morin & Bengio, 2005), often used with neural networks to speed up computations in multi-class classification with large output spaces. For example, it is commonly applied in natural language processing problems such as language modeling (Mikolov et al., 2013). To adapt HSM to *extreme multi-label classification* (XMLC), several very popular tools, such as FASTTEXT (Joulin et al., 2016) and LEARNED TREE (Jernite et al., 2017), apply the *pick-one-label* heuristic. As the name suggests, this heuristic randomly picks one of the labels from a multi-label training example and treats the example as a multi-class one.

In this work, we exhaustively investigate the multi-label extensions of HSM. First, we show that the pick-one-label strategy does not lead to a proper generalization of HSM for multi-label setting. More precisely, we prove that using the pick-one-label reduction one cannot expect any multi-class learner to achieve zero regret in terms of marginal probability estimation and maximization of precision@$k$. As a remedy to this issue, we are going to revisit *probabilistic label trees* (PLTs) (Jasinska et al., 2016) that have been recently introduced for solving XMLC problems. We show that PLTs are a theoretically motivated generalization of HSM to multi-label classification, that is, 1) PLTs and HSM are identical in multi-class case, and 2) a PLT model can get *zero regret* (i.e., it is *consistent*) in terms of marginal probability estimation and precision@$k$ in the multi-label setting.

Beside our theoretical findings, we provide an efficient implementation of PLTs, referred to as XT, that we build upon FASTTEXT. The comprehensive empirical evaluation shows that it gets significantly better results than the original FASTTEXT, LEARNED TREE, and XML-CNN, a specifically designed deep network for XMLC problems. XT also achieves competitive results to other state-of-the-art approaches, being very efficient in model size and prediction time, particularly in the online setting.

This paper is organized as follows. First we discuss the related work and situate our approach in the context. In Section 3 we formally state the XMLC problem and present some useful theoretical insights. Next, we briefly introduce the HSM approach, and in Section 5 we show theoretical results concerning the pick-one-label heuristic. Section 6 formally introduces PLTs and presents the main theoretical results concerning them and their relation to HSM. Section 7 provides implementation details of PLTs. The experimental results are presented in Section 8. Finally we make concluding remarks.

## 2    Related work

Historically, problems with a large number of labels were usually solved by nearest neighbor or decision tree methods. Some of today's algorithms are still based on these classical approaches, significantly extending them by a number of new tricks. If the label space is of moderate size (like a few thousands of labels) then an independent model can be trained for each class. This is the so-called 1-VS-ALL approach. Unfortunately, it scales linearly with the number of labels, which is too costly for many applications. The extreme classification algorithms try to improve over this approach by following different paradigms such as sparsity of labels (Yen et al., 2017; Babbar & Schölkopf, 2017), low-rank approximation (Mineiro & Karampatziakis, 2015; Yu et al., 2014; Bhatia et al., 2015), tree-based search (Prabhu & Varma, 2014; Choromanska & Langford, 2015), or label filtering (Vijayanarasimhan et al., 2014; Shrivastava & Li, 2015; Niculescu-Mizil & Abbasnejad, 2017).

In this paper we focus on tree-based algorithms, therefore we discuss them here in more detail. There are two distinct types of these algorithms: decision trees and label trees. The former type follows the idea of classical decision trees. However, the direct use of the classic algorithms can be very costly (Agrawal et al., 2013). Therefore, the FASTXML algorithm (Prabhu & Varma, 2014) tackles the problem in a slightly different way. It uses sparse linear classifiers in internal tree nodes to split the feature space. Each linear classifier is trained on two classes that are formed in a random way first and then reshaped by optimizing the normalized discounted cumulative gain. To improve the overall accuracy FASTXML uses an ensemble of trees. This algorithm, like many other decision tree methods, works in a batch mode. Choromanska & Langford (2015) have succeeded to introduce a fully online decision tree algorithm that also uses linear classifiers in internal nodes of the tree.

In label trees each label corresponds to one and only one path from the root to a leaf. Besides PLTs and HSM, there exist several other instances of this approach, for example, filter trees (Beygelzimer et al., 2009b; Li & Lin, 2014) or label embedding trees (Bengio et al., 2010). It is also worth to underline that algorithms similar to HSM have been introduced independently in many different research fields, such as nested dichotomies (Fox, 1997) in statistics, conditional probability estimation trees (Beygelzimer et al., 2009a) in multi-class classification, multi-stage classifiers (Kurzynski, 1988) in pattern recognition, and probabilistic classifier chains (Dembczynski et al., 2010) in multi-label classification under the subset 0/1 loss. All these methods have been jointly analyzed in (Dembczynski et al., 2016).

A still open problem in label tree approaches is the tree structure learning. FASTTEXT (Joulin et al., 2016) uses HSM with a Huffman tree built on the label frequencies. Jernite et al. (2017)

have introduced a new algorithm, called LEARNED TREE, which combines HSM with a specific hierarchical clustering that reassigns labels to paths in the tree in a semi-online manner. Prabhu et al. (2018) follows another approach in which a model similar to PLTs is trained in a batch mode and a tree is built by using recursively balanced k-means over the label profiles. In Section 7 we discuss this approach in more detail.

The HSM model is often used as an output layer in neural networks. The FASTTEXT implementation can also be viewed as a shallow architecture with one hidden layer that represents instances as averaged feature (i.e., word) vectors. Another neural network-based model designed for XMLC has been introduced in (Liu et al., 2017). This model, referred to as XML-CNN, uses a complex convolutional deep network with a narrow last layer to make it work with large output spaces. As we show in the experimental part, this quite expensive architecture gets inferior results in comparison to our PLTs built upon FASTTEXT.

## 3   Problem statement

Let $\mathcal{X}$ denote an instance space, and let $\mathcal{L} = \{1, \ldots, m\}$ be a finite set of $m$ class labels. We assume that an instance $\boldsymbol{x} \in \mathcal{X}$ is associated with a subset of labels $\mathcal{L}_{\boldsymbol{x}} \in 2^{\mathcal{L}}$ (the subset can be empty); this subset is often called a set of relevant labels, while the complement $\mathcal{L} \backslash \mathcal{L}_{\boldsymbol{x}}$ is considered as irrelevant for $\boldsymbol{x}$. We assume $m$ to be a large number (e.g., $\geq 10^5$), but the size of the set of relevant labels $\mathcal{L}_{\boldsymbol{x}}$ is much smaller than $m$, i.e., $|\mathcal{L}_{\boldsymbol{x}}| \ll m$. We identify a set $\mathcal{L}_{\boldsymbol{x}}$ of relevant labels with a binary (sparse) vector $\boldsymbol{y} = (y_1, y_2, \ldots, y_m)$, in which $y_j = 1 \Leftrightarrow j \in \mathcal{L}_{\boldsymbol{x}}$. By $\mathcal{Y} = \{0, 1\}^m$ we denote a set of all possible label vectors. We assume that observations $(\boldsymbol{x}, \boldsymbol{y})$ are generated independently and identically according to the probability distribution $\mathbf{P}(\boldsymbol{X} = \boldsymbol{x}, \boldsymbol{Y} = \boldsymbol{y})$ (denoted later by $\mathbf{P}(\boldsymbol{x}, \boldsymbol{y})$) defined on $\mathcal{X} \times \mathcal{Y}$.

The problem of XMLC can be defined as finding a *classifier* $\boldsymbol{h}(\boldsymbol{x}) = (h_1(\boldsymbol{x}), h_2(\boldsymbol{x}), \ldots, h_m(\boldsymbol{x}))$, which in general can be defined as a mapping $\mathcal{X} \to \mathcal{R}^m$, that minimizes the *expected loss* (or *risk*):

$$L_\ell(\boldsymbol{h}) = \mathbb{E}_{(\boldsymbol{x}, \boldsymbol{y}) \sim \mathbf{P}(\boldsymbol{x}, \boldsymbol{y})}(\ell(\boldsymbol{y}, \boldsymbol{h}(\boldsymbol{x}))),$$

where $\ell(\boldsymbol{y}, \hat{\boldsymbol{y}})$ is the *(task) loss*. The optimal classifier, the so-called *Bayes classifier*, for a given loss function $\ell$ is:

$$\boldsymbol{h}_\ell^* = \arg\min_{\boldsymbol{h}} L_\ell(\boldsymbol{h}).$$

The *regret* of a classifier $\boldsymbol{h}$ with respect to $\ell$ is defined as:

$$\mathrm{reg}_\ell(\boldsymbol{h}) = L_\ell(\boldsymbol{h}) - L_\ell(\boldsymbol{h}_\ell^*) = L_\ell(\boldsymbol{h}) - L_\ell^*.$$

The regret quantifies the suboptimality of $\boldsymbol{h}$ compared to the optimal classifier $\boldsymbol{h}^*$. The goal could be then defined as finding $\boldsymbol{h}$ with a small regret, ideally equal to zero.

In the following, we aim at estimating the marginal probabilities $\eta_j(\boldsymbol{x}) = \mathbf{P}(y_j = 1 \,|\, \boldsymbol{x})$. As we will show below, marginal probabilities are a key element to optimally solve extreme classification for many performance measures, like Hamming loss, macro-F measure, and precision@$k$. To obtain the marginal probability estimates one can use the label-wise log loss as a surrogate:

$$\ell_{\log}(\boldsymbol{y}, \boldsymbol{h}(\boldsymbol{x})) = \sum_{j=1}^m \ell_{\log}(y_j, h_j(\boldsymbol{x})) = \sum_{j=1}^m (y_j \log(h_j(\boldsymbol{x})) + (1 - y_j) \log(1 - h_j(\boldsymbol{x}))).$$

Then the expected label-wise log loss for a single $\boldsymbol{x}$ (i.e., the so-called *conditional risk*) is:

$$\mathbb{E}_{\boldsymbol{y}} \ell_{\log}(\boldsymbol{y}, \boldsymbol{h}(\boldsymbol{x})) = \sum_{j=1}^m \mathbb{E}_{\boldsymbol{y}} \ell_{\log}(y_j, h_j(\boldsymbol{x})) = \sum_{j=1}^m L_{\log}(h_j(\boldsymbol{x}) \,|\, \boldsymbol{x}).$$

Therefore, it is easy to see that the pointwise optimal prediction for the $j$-th label is given by:

$$h_j^*(\boldsymbol{x}) = \arg\min_{h} L_{\log}(h_j(\boldsymbol{x}) \,|\, \boldsymbol{x}) = \eta_j(\boldsymbol{x}).$$

As shown in (Dembczynski et al., 2010), the Hamming loss is minimized by $h_j^*(\boldsymbol{x}) = [\![\eta_j(\boldsymbol{x}) > 0.5]\!]$. For the macro F-measure it suffices in turn to find an optimal threshold on marginal probabilities for

each label separately as proven in (Ye et al., 2012; Narasimhan et al., 2014; Jasinska et al., 2016; Dembczynski et al., 2017). In the following, we will show a similar result for precision@$k$ which has become a standard measure in extreme classification (although it is also often criticized, as it favors the most frequent labels).

Precision@$k$ can be formally defined as:

$$\text{precision@}k(\boldsymbol{y}, \boldsymbol{x}, \boldsymbol{h}) = \frac{1}{k} \sum_{j \in \hat{\mathcal{Y}}_k} [\![ y_j = 1 ]\!], \tag{1}$$

where $\hat{\mathcal{Y}}_k$ is a set of $k$ labels predicted by $\boldsymbol{h}$ for $\boldsymbol{x}$. To be consistent with the former discussion, let us define a loss function for precision@$k$ as $\ell_{p@k} = 1 - \text{precision@}k$. The conditional risk is then:[1]

$$L_{p@k}(\boldsymbol{h} \,|\, \boldsymbol{x}) = \mathbb{E}_{\boldsymbol{y}} \ell_{p@k}(\boldsymbol{y}, \boldsymbol{x}, \boldsymbol{h}) = 1 - \frac{1}{k} \sum_{j \in \hat{\mathcal{Y}}_k} \eta_j(\boldsymbol{x}) \,.$$

The above result shows that the optimal strategy for precision@$k$ is to predict $k$ labels with the highest marginal probabilities $\eta_j(\boldsymbol{x})$. As the main theoretical result given in this paper is a regret bound for precision@$k$, let us define here the conditional regret for this metric:

$$\text{reg}_{p@k}(\boldsymbol{h} \,|\, \boldsymbol{x}) = \frac{1}{k} \sum_{i \in \mathcal{Y}_k} \eta_i(\boldsymbol{x}) - \frac{1}{k} \sum_{j \in \hat{\mathcal{Y}}_k} \eta_j(\boldsymbol{x}) \,,$$

where $\mathcal{Y}_k$ is a set containing the top $k$ labels with respect to the true marginal probabilities.

From the above results, we see that estimation of marginal probabilities is crucial for XMLC problems. To obtain these probabilities we can use the vanilla 1-VS-ALL approach trained with the label-wise log loss. Unfortunately, 1-VS-ALL is too costly in the extreme setting. In the following sections, we discuss an alternative approach based on the label trees that estimates the marginal probabilities with the competitive accuracy, but in a much more efficient way.

## 4  Hierarchical softmax approaches

Hierarchical softmax (HSM) is designed for multi-class classification. Using our notation, for multi-class problems we have $\sum_{i=1}^m y_i = 1$, i.e., there is one and only one label assigned to an instance $(\boldsymbol{x}, \boldsymbol{y})$. The marginal probabilities $\eta_j(\boldsymbol{x})$ in this case sum up to 1.

The HSM classifier $\boldsymbol{h}(\boldsymbol{x})$ takes a form of a label tree. We encode all labels from $\mathcal{L}$ using a prefix code. Any such code can be given in a form of a tree in which a path from the root to a leaf node corresponds to a code word. Under the coding, each label $y_j = 1$ is uniquely represented by a code word $\boldsymbol{z} = (z_1, \ldots, z_l) \in \mathcal{C}$, where $l$ is the length of the code word and $\mathcal{C}$ is a set of all code words. For $z_i \in \{0, 1\}$, the code and the label tree are binary. In general, the code alphabet can contain more than two symbols. Furthermore, $z_i$s can take values from different sets of symbols depending on the previous values in the code word. In other words, the code can result with nodes of a different arity even in the same tree, like in (Grave et al., 2017) and (Prabhu et al., 2018). We will briefly discuss different tree structures in Section 7.

A tree node can be uniquely identified by the partial code word $\boldsymbol{z}^i = (z_1, \ldots, z_i)$. We denote the root node by $\boldsymbol{z}^0$, which is an empty vector (without any elements). The probability of a given label is determined by a sequence of decisions made by node classifiers that predict subsequent values of the code word. By using the chain rule of probability, we obtain:

$$\eta_j(\boldsymbol{x}) = \mathbf{P}(y_j = 1 \,|\, \boldsymbol{x}) = \mathbf{P}(\boldsymbol{z} \,|\, \boldsymbol{x}) = \prod_{i=1}^l \mathbf{P}(z_i \,|\, \boldsymbol{z}^{i-1}, \boldsymbol{x}) \,.$$

By using logistic loss and a linear model $f_{\boldsymbol{z}^i}(\boldsymbol{x})$ in each node $\boldsymbol{z}^i$ for estimating $\mathbf{P}(z_i \,|\, \boldsymbol{z}^{i-1}, \boldsymbol{x})$, we obtain the popular formulation of HSM. Let us notice that since we deal here with a multi-class distribution, we have that:

$$\sum_c \mathbf{P}(z_i = c \,|\, \boldsymbol{z}^{i-1}, \boldsymbol{x}) = 1 \,. \tag{2}$$

Because of this normalization, we can assume that a multi-class (or binary in the case of binary trees) classifier is situated in all internal nodes and there are no classifiers in the leaves of the tree. Alternatively, we can assume that each node, except the root, is associated with a binary classifier that estimates $\mathbf{P}(z_i = c \,|\, \boldsymbol{z}^{i-1}, \boldsymbol{x})$, but then the additional normalization (2) has to be performed. This alternative formulation is important for the multi-label extension of HSM discussed in Section 6. In either way, learning of the node classifiers can be performed simultaneously as independent tasks.

Note that estimate $\hat{\eta}_j(\boldsymbol{x})$ of the probability of label $j$ can be easily obtained by traversing the tree along the path indicated by the code of the label. Unfortunately, the task of predicting top $k$ labels is more involved as it requires searching over the tree. Popular solutions are beam search (Kumar et al., 2013; Prabhu et al., 2018), uniform-cost search (Joulin et al., 2016), and its approximate variant (Dembczynski et al., 2012, 2016).

## 5 Suboptimality of HSM for multi-label classification

To deal with multi-label problems, some popular tools, such as FASTTEXT (Joulin et al., 2016) and its extension LEARNED TREE (Jernite et al., 2017), apply HSM with the pick-one-label heuristic which randomly picks one of the positive labels from a given training instance. The resulting instance is then treated as a multi-class instance. During prediction, the heuristic returns a multi-class distribution and the $k$ most probable labels. We show below that this specific reduction of the multi-label problem to multi-class classification is not consistent in general.

Since the probability of picking a label $j$ from $\boldsymbol{y}$ is equal to $y_j / \sum_{j'=1}^{m} y_{j'}$, the pick-one-label heuristic maps the multi-label distribution to a multi-class distribution in the following way:

$$\eta_j'(\boldsymbol{x}) = \mathbf{P}'(y_j = 1 \,|\, \boldsymbol{x}) = \sum_{\boldsymbol{y} \in \mathcal{Y}} \frac{y_j}{\sum_{j'=1}^{m} y_{j'}} \mathbf{P}(\boldsymbol{y} \,|\, \boldsymbol{x}) \tag{3}$$

It can be easily checked that the resulting $\eta_j'(\boldsymbol{x})$ form a multi-class distribution as the probabilities sum up to 1. It is obvious that that the heuristic changes the marginal probabilities of labels, unless the initial distribution is multi-class. Therefore this method cannot lead to consistent classifiers in terms of estimating $\eta_j(\boldsymbol{x})$. As we show below, it is also not consistent for precision@$k$ in general.

**Proposition 1.** *A classifier $\boldsymbol{h}$ such that $h_j(\boldsymbol{x}) = \eta_j'(\boldsymbol{x})$ for all $j \in \{1, \ldots, m\}$ has in general a non-zero regret in terms of precision@$k$.*

*Proof.* We prove the proposition by giving a simple counterexample. Consider the following conditional distribution for some $\boldsymbol{x}$:

$$\mathbf{P}(\boldsymbol{y} = (1,0,0) \,|\, \boldsymbol{x}) = 0.1, \quad \mathbf{P}(\boldsymbol{y} = (1,1,0) \,|\, \boldsymbol{x}) = 0.5, \quad \mathbf{P}(\boldsymbol{y} = (0,0,1) \,|\, \boldsymbol{x}) = 0.4.$$

The optimal top 1 prediction for this example is obviously label 1, since the marginal probabilities are $\eta_1(\boldsymbol{x}) = 0.6, \eta_2(\boldsymbol{x}) = 0.5, \eta_3(\boldsymbol{x}) = 0.4$. However, the pick-one-label heuristic will transform the original distribution to the following one: $\eta_1'(\boldsymbol{x}) = 0.35, \eta_2'(\boldsymbol{x}) = 0.25, \eta_3'(\boldsymbol{x}) = 0.4$. The predicted top label will be then label 3, giving the regret of 0.2 for precision@1. $\square$

The proposition shows that the heuristic is in general inconsistent for precision@$k$. Interestingly, the situation changes when the labels are conditionally independent, i.e., $\mathbf{P}(\boldsymbol{y} \,|\, \boldsymbol{x}) = \prod_{j=1}^{m} \mathbf{P}(y_i \,|\, \boldsymbol{x})$.

**Proposition 2.** *Given conditionally independent labels, a classifier $\boldsymbol{h}$ such that $h_j(\boldsymbol{x}) = \eta_j'(\boldsymbol{x})$ for all $j \in \{1, \ldots, m\}$ has zero regret in terms of the precision@$k$ loss.*

*Proof.* We show here only a sketch of the proof. The full proof is given in Appendix B. To prove the theorem, it is enough to show that in the case of conditionally independent labels the pick-one-label heuristic does not change the order of marginal probabilities. Let $y_i$ and $y_j$ be so that $\mathbf{P}(y_i = 1 \,|\, \boldsymbol{x}) \geq \mathbf{P}(y_j = 1 \,|\, \boldsymbol{x})$. Then in the summation over all $\boldsymbol{y}$s in (3), we are interested in four different subsets of $\mathcal{Y}$, $S_{i,j}^{u,w} = \{\boldsymbol{y} \in \mathcal{Y} : y_i = u \wedge y_j = w\}$, where $u, w \in \{0, 1\}$. Remark that during mapping none of $\boldsymbol{y} \in S_{i,j}^{0,0}$ plays any role, and for each $\boldsymbol{y} \in S_{i,j}^{1,1}$, the value of $y_t / (\sum_{t'=1}^{m} y_{t'}) \times \mathbf{P}(\boldsymbol{y} \,|\, \boldsymbol{x})$, for $t \in \{i, j\}$, is the same for both $y_i$ and $y_j$. Now, let $\boldsymbol{y}' \in S_{i,j}^{1,0}$ and $\boldsymbol{y}'' \in S_{i,j}^{0,1}$ be the same on all elements except the $i$-th and the $j$-th one. Then, because

of the label independence and the assumption that $\mathbf{P}(y_i = 1 \,|\, \boldsymbol{x}) \geq \mathbf{P}(y_j = 1 \,|\, \boldsymbol{x})$, we have $\mathbf{P}(\boldsymbol{y}' \,|\, \boldsymbol{x}) \geq \mathbf{P}(\boldsymbol{y}'' \,|\, \boldsymbol{x})$. Therefore, after mapping we obtain $\eta_i'(\boldsymbol{x}) \geq \eta_j'(\boldsymbol{x})$. Thus, for independent labels, the pick-one-label heuristic is consistent for precision@$k$. □

## 6   Probabilistic label trees

The section above has revealed that HSM cannot be properly adapted to multi-label problems by the pick-one-label heuristic. There is, however, a different way to generalize HSM to obtain no-regret estimates of marginal probabilities $\eta_j(\boldsymbol{x})$. The probabilistic label trees (PLTs) (Jasinska et al., 2016) can be derived in the following way. Let us encode $y_j = 1$ by a slightly extended code $\boldsymbol{z} = (1, z_1, \dots, z_l)$ in comparison to HSM. The new code gets 1 at the zero position what corresponds to a question whether there exists at least one label assigned to the example. As before, each node is uniquely identified by a partial code $\boldsymbol{z}^i$ which says that there is at least one positive label in a subtree rooted in that node. It can be easily shown by the chain rule of probability that the marginal probabilities can be expressed in the following way:

$$\eta_j(\boldsymbol{x}) = \mathbf{P}(\boldsymbol{z} \,|\, \boldsymbol{x}) = \prod_{i=0}^{l} \mathbf{P}(z_i \,|\, \boldsymbol{z}^{i-1}, \boldsymbol{x}) \,. \tag{4}$$

The difference to HSM is the probability $\mathbf{P}(z_0 = 1 \,|\, \boldsymbol{x})$ in the chain and a different normalization, i.e.:

$$\sum_c \mathbf{P}(z_i = c \,|\, \boldsymbol{z}^{i-1}, \boldsymbol{x}) \geq 1 \,. \tag{5}$$

Only for $z_0$ we have $\mathbf{P}(z_0 = 1 \,|\, \boldsymbol{x}) + \mathbf{P}(z_0 = 0 \,|\, \boldsymbol{x}) = 1$. Because of (5), the binary models that estimate $\mathbf{P}(z_i = c \,|\, \boldsymbol{z}^{i-1}, \boldsymbol{x})$ (against $\mathbf{P}(z_i \neq c \,|\, \boldsymbol{z}^{i-1}, \boldsymbol{x})$) are situated in all nodes of the tree (i.e., also in the leaves). The models can be trained independently as before for HSM. Only during prediction, one can re-calibrate the estimates when (5) is not satisfied, for example, by normalizing them to sum up to 1. It can be easily noticed that for a multi-class distribution, the resulting model of PLTs boils down to HSM, since $\mathbf{P}(z_0 = 1 \,|\, \boldsymbol{x})$ is always equal 1, and in addition, normalization (5) will take the form of (2). In Appendix D we additionally present the pseudocode of training and predicting with PLTs.

Next, we show that the PLT model obeys strong theoretical guarantees. Let us first revise the result from (Jasinska et al., 2016) that relates the absolute difference between the true and the estimated marginal probability of label $j$, $|\eta_j(\boldsymbol{x}) - \hat{\eta}_j(\boldsymbol{x})|$, to the surrogate loss $\ell$ used to train node classifiers $f_{\boldsymbol{z}^i}$. It is assumed here that $\ell$ is a strongly proper composite loss (e.g, logistic, exponential, or squared loss) characterized by a constant $\lambda$ (e.g. $\lambda = 4$ for logistic loss).[2]

**Theorem 1.** *For any distribution $\mathbf{P}$ and internal node classifiers $f_{\boldsymbol{z}^i}$, the following holds:*

$$|\eta_j(\boldsymbol{x}) - \hat{\eta}_j(\boldsymbol{x})| \leq \sum_{i=0}^{l} \mathbf{P}(\boldsymbol{z}^{i-1} \,|\, \boldsymbol{x}) \sqrt{\frac{2}{\lambda}} \sqrt{\mathrm{reg}_\ell(f_{\boldsymbol{z}^i} \,|\, \boldsymbol{z}^{i-1}, \boldsymbol{x})} \,,$$

*where $\mathrm{reg}_\ell(f_{\boldsymbol{z}^i} \,|\, \boldsymbol{z}^{i-1}, \boldsymbol{x})$ is a binary classification regret for a strongly proper composite loss $\ell$ and $\lambda$ is a constant specific for loss $\ell$.*

Due to filtering of the distribution imposed by the PLT, the regret $\mathrm{reg}_\ell(f_{\boldsymbol{z}^i} \,|\, \boldsymbol{z}^{i-1}, \boldsymbol{x})$ of a classifier $f_{\boldsymbol{z}^i}$ exists only for $\boldsymbol{x}$ such that $\mathbf{P}(\boldsymbol{z}^{i-1} \,|\, \boldsymbol{x}) > 0$, therefore we condition the regret not only on $\boldsymbol{x}$, but also on $\boldsymbol{z}^{i-1}$. The above result shows that the absolute error of estimating the marginal probability of label $j$ can be upper bounded by the regret of the node classifiers on the corresponding path from the root to a leaf. The proof of Theorem 1 is given in Appendix A. Moreover, for zero-regret (i.e., optimal) node classifiers we obtain an optimal multi-label classifier in terms of estimation of marginal probabilities $\eta_j(\boldsymbol{x})$. This result can be further extended for precision@$k$.

**Theorem 2.** *For any distribution $\mathbf{P}$ and classifier $\boldsymbol{h}$ delivering estimates $\hat{\eta}_j(\boldsymbol{x})$ of the marginal probabilities of labels, the following holds:*

$$\mathrm{reg}_{p@k}(\boldsymbol{h} \,|\, \boldsymbol{x}) = \frac{1}{k} \sum_{i \in \mathcal{Y}_k} \eta_i(\boldsymbol{x}) - \frac{1}{k} \sum_{j \in \hat{\mathcal{Y}}_k} \eta_j(\boldsymbol{x}) \leq 2 \max_l |\eta_l(\boldsymbol{x}) - \hat{\eta}_l(\boldsymbol{x})|$$

The proof is based on adding and subtracting the following terms $\frac{1}{k}\sum_{i \in \mathcal{Y}_k} \hat{\eta}_i(\boldsymbol{x})$ and $\frac{1}{k}\sum_{j \in \hat{\mathcal{Y}}_k} \hat{\eta}_j(\boldsymbol{x})$ to the regret (a detailed proof is given in Appendix A). By getting together both theorems we get an upper bound of the precision@$k$ regret expressed in terms of the regret of the node classifiers. Again, for the zero-regret node classifiers, we get optimal solution in terms of precision@$k$.

# 7    Implementation details of PLTs

Given the tree structure, the node classifiers of PLTs can be trained as logistic regression either in online (Jasinska et al., 2016) or batch mode (Prabhu et al., 2018). Both training modes have their pros and cons, but the online implementation gives a possibility of learning more complex representation of input instances. The above cited implementations are both based on sparse representation, given either in a form of a bag-of-words or its TF-IDF variant. We opt here for training a PLT in the online mode along with the dense representation. We build our implementation upon FASTTEXT and refer to it as XT which stands for EXTREMETEXT.[3] In this way, we succeeded to obtain a very powerful and compressed model. The small dense models are important for fast online prediction as they do not need too much resources. The sparse models, in turn, can be slow and expensive in terms of memory usage as they need to decompress the node models to work fast. Remark also that, in general, PLTs can be used as an output layer of any neural network architecture (also that one used in XML-CNN (Yen et al., 2017)) to speed up training and prediction time.

In contrast to the original implementation of FASTTEXT, we use L2 regularization for all parameters of the model. To obtain representation of input instances we do not compute simple averages of the feature vectors, but use weights proportional to the TF-IDF scores of features. The competitive results can be obtained with feature and instance vectors of size 500. If a node classification task contains only positive instances, we use a constant classifier predicting 1 without any training. The training of PLT in either mode, online or batch, can be easily parallelized as each node classifier can be trained in isolation from the other classifiers. In our current implementation, however, we follow the parallelization on the level of training and test instances as in original FASTTEXT.

Our implementation, because of the additional use of the L2 regularization, has more parameters than original FASTTEXT. We have found, however, that our model is remarkably robust for the hyperparameter selection, since it achieves close to optimal performance for a large set of hyperparameters that is in the vicinity of the optimal one. Moreover, the optimal hyperparameters are close to each other across all datasets. We report more information about the hyperparameter selection in Appendix E.4.

The tree structure of a PLT is a crucial modeling decision. The vanishing regret for probability estimates and precision@$k$ holds regardless of the tree structure (see Theorem 1 and 2), however, this theory requires the regret of the node classifiers also to vanish. In practice, we can only estimate the conditional probabilities in the nodes, therefore the tree structure does indeed matter as it affects the difficulty of the node learning problems. The original PLT paper (Jasinska et al., 2016) uses simple complete trees with labels assigned to leaves according to their frequencies. Another option, routinely used in HSM (Joulin et al., 2016), is the Huffman tree built over the label frequencies. Such tree takes into account the computational complexity by putting the most frequent labels close to the root. This approach has been further extended to optimize GPU operations in (Grave et al., 2017). Unfortunately, it ignores the statistical properties of the tree structure. Furthermore, for multi-label case the Huffman tree is no longer optimal even in terms of computational cost as we show it in Appendix C. There exist, however, other methods that focus on building a tree with high overall accuracy (Tagami, 2017; Prabhu et al., 2018). In our work, we follow the later approach, which performs a simple top-down hierarchical clustering. Each label in this approach is represented by a profile vector being an average of the training vectors tagged by this label. Then the profile vectors are clustered using balanced k-means which divides the labels into two or more clusters with approximately the same size. This procedure is then repeated recursively until the clusters are smaller than a given value (for example, 100). The nodes of the resulting tree are then of different arities. The internal nodes up to the leaves' parent nodes have $k$ children, but the leaves' parent nodes are usually of higher arity. Thanks to this clustering, similar labels are close to each other in the tree. Moreover, the tree is balanced, so its depth is logarithmic in terms of the number of labels.

# 8 Empirical results

We carried out three sets of experiments. In the first, we compare exhaustively the performance of PLTs and HSM on synthetic and benchmark data. Due to lack of space, the results are deferred to Appendix E.1 and E.2. The results on synthetic data confirm our theoretical findings: the models are the same in the case of multi-class data, the performance of HSM and PLTs is on par using multi-label data with independent labels, and PLTs significantly outperform HSM on multi-label data with conditionally dependent labels. The results on the benchmark data clearly indicate the better performance of PLTs over HSM.

In the second experiment, we compare XT, the variant of PLTs discussed in the previous section, to the state-of-the-art algorithms on five benchmark datasets taken from XMLC repository,[4] and their text equivalents, by courtesy of Liu et al. (2017). We compare the models in terms of precision@$\{1, 3, 5\}$, model size, training and test time. The competitors for our XT are original FASTTEXT, its variant LEARNED TREE, a PLT-like batch learning algorithm PARABEL (we use the variant that uses a single tree instead of an ensemble), a XMLC-designed convolutional deep network XML-CNN, a decision tree ensemble FASTXML, and two 1-vs-All approaches tailored to XMLC problems, PPD-SPARSE and DISMEC. The hyperparameters of the models have been tuned using grid search. The range of the hyperparameters is reported in E.4.

The results presented in Table 1 demonstrate that XT outperforms the HSM approaches with the pick-one-label heuristic, namely FASTTEXT and LEARNED TREE, with a large margin. This proves the superiority of PLTs as the proper generalization of HSM to multi-label setting. In all the above methods we use vectors of length 500 and we tune the other hyperparameters appropriately for a fair comparison.

Moreover, XT scales well to extreme datasets achieving performance close to the state-of-the-art, being at the same time 10000x and 100x faster compared to DISMEC and PPDSPARSE during prediction. XT always responds below 2ms, what makes it a competitive alternative for an online setting. XT is also close to PARABEL in terms of performance. However, the reported times and model sizes of PARABEL are given for the batch prediction. The prediction times seem to be faster, but PARABEL needs to decompress the model during prediction, what makes it less suitable for online prediction. It is only efficient when the batches are sufficiently large. Finally, we would like to underline that XT outperforms XML-CNN, the more complex neural network, in terms of predictive performance with computational costs that are an order of magnitude smaller. Moreover, XML-CNN requires pretrained embedding vectors, whereas XT can be used with random initialization.

In the third experiment we perform an ablation analysis in which we compare different components of the XT algorithm. We analyze the influence of the Huffman tree vs. top-down clustering, the simple averaging of features vectors vs. the TF-IDF-based weighting, and no regularization vs. L2 regularization. Figure 1 clearly shows that the components need to be combined together to obtain the best results. The best combination uses top-down clustering, TF-IDF-based weighting, and L2 regularization, while top-down clustering alone gets worse results than Huffman trees with TF-IDF-based weighting and L2 regularization. In Appendix E.3 we give more detailed results of the ablation analysis performed on a larger spectrum of benchmark datasets.

# 9 Conclusions

In this paper we have proven that probabilistic label trees (PLTs) are no-regret generalization of HSM to the multi-label setting. Our main theoretical contribution is the precision@$k$ regret bound for PLTs. Moreover, we have shown that the pick-one-label heuristic commonly-used with HSM in multi-label problems leads to inconsistent results in terms of marginal probability estimation and precision@$k$. Our implementation of PLTs referred to as XT, built upon FASTTEXT, gets state-of-the-art results, being significantly better than the original FASTTEXT, LEARNED TREE, and XML-CNN. The XT results are also close to the best known ones that are obtained by expensive 1-vs-All approaches, such as PPDSPARSE and DISMEC, and outperforms the other tree-based methods on many benchmarks. Our online variant has the advantage of producing very often much smaller models that can be efficiently used in fast online prediction.

Table 1: Precision@$k$ scores with $k = \{1, 3, 5\}$ and statistics of FASTXML, PPDSPARSE, DISMEC, PARABEL (with 1 tree), FASTTEXT (FT), LEARNED TREE (LT), EXTREMETEXT (XT) and XML-CNN methods. Notation: $N$ – number of samples, $T$ – CPU time, $m$ – number of labels, $d$ – number of features, $*$ – result of offline prediction, $\star$ – calculated on GPU, † – not reported by authors, ‡ – cannot be calculated due to lack of a text version of a dataset.

| Dataset | Metrics | FASTXML | PPDSPARSE | DISMEC | FT | LT | XT | PARABEL | XML-CNN |
|---|---|---|---|---|---|---|---|---|---|
| **Wiki-30K** | P@1 | 82.03 | 73.80 | 85.20 | 80.78 | 80.85 | **85.23** | 83.77 | 82.78 |
| $N_{train} = 14146$ | P@3 | 67.47 | 60.90 | **74.60** | 50.46 | 50.59 | 73.18 | 71.96 | 66.34 |
| $N_{test} = 6616$ | P@5 | 57.76 | 50.40 | **65.90** | 36.79 | 37.68 | 63.39 | 62.44 | 56.23 |
| $d = 101938$ | $T_{train}$ | 16m | † | † | 10m | 12m | 18m | **5m** | 88m★ |
| $m = 30938$ | $T_{test}/N_{test}$ | 3.00ms | † | † | 1.88ms | 10.09ms | **0.83ms** | 1.63ms* | 1.39ms★ |
| | model size | 354M | † | † | 513M | 513M | **259M** | 109M* | ★ |
| **Delicious-200K** | P@1 | 42.81 | 45.05 | 44.71 | 42.22 | 42.71 | **47.85** | 43.32 | ‡ |
| $N_{train} = 196606$ | P@3 | 38.76 | 38.34 | 38.08 | 37.90 | 36.27 | **42.08** | 38.49 | ‡ |
| $N_{test} = 100095$ | P@5 | 36.34 | 34.90 | 34.7 | 35.05 | 33.43 | **39.13** | 35.83 | ‡ |
| $d = 782585$ | $T_{train}$ | 458m | 4781m | 1080h | 271m | 563m | 502m | 105m | ‡ |
| $m = 205443$ | $T_{test}/N_{test}$ | 4.86ms | 275ms | 5m | 1.97ms | 1.98ms | **1.41ms** | 1.31ms* | ‡ |
| | model size | 15.4G | 9.4G | 18.0G | 9.0G | 9.0G | **1.9G** | 1.8G* | ‡ |
| **WikiLSHTC** | P@1 | 49.35 | 64.08 | **64.94** | 41.13 | 50.15 | 58.73 | 61.53 | ‡ |
| $N_{train} = 1778351$ | P@3 | 32.69 | 41.26 | **42.71** | 24.09 | 31.95 | 39.24 | 40.07 | ‡ |
| $N_{test} = 587084$ | P@5 | 24.03 | 30.12 | **31.5** | 17.44 | 23.59 | 29.26 | 29.25 | ‡ |
| $d = 617899$ | $T_{train}$ | 724m | 236m | 750h | 207 | 212m | 550m | **34m** | ‡ |
| $m = 325056$ | $T_{test}/N_{test}$ | 2.17ms | 37.76ms | 43m | 1.25ms | 4.76ms | **0.81ms** | 0.92ms* | ‡ |
| | model size | 9.3G | 5.2G | 3.8G | 6.5G | 6.5G | **3.3G** | 1.1G* | ‡ |
| **Wiki-500K** | P@1 | 54.10 | 70.16 | **70.20** | 32.73 | 37.18 | 64.48 | 66.12 | 59.85 |
| $N_{train} = 1813391$ | P@3 | 29.45 | 50.57 | **50.60** | 19.02 | 21.62 | 45.84 | 47.02 | 39.28 |
| $N_{test} = 783743$ | P@5 | 21.21 | 39.66 | **39.70** | 14.46 | 16.01 | 35.46 | 36.45 | 29.81 |
| $d = 2381304$ | $T_{train}$ | 3214m | 1771m | 7495h | 496m | 531m | 1253m | **168m** | 7032m★ |
| $m = 501070$ | $T_{test}/N_{test}$ | 8.03ms | 113.70ms | 155m | 2.05ms | 6.43ms | **1.07ms** | 4.68ms* | 21.06ms★ |
| | model size | 63G | 3.4G | 14.7G | 11G | 11G | 5.5G | **2.0G*** | 3.7G★ |
| **Amazon-670K** | P@1 | 34.24 | 45.32 | **45.37** | 25.47 | 27.67 | 39.90 | 41.59 | 35.39 |
| $N_{train} = 490449$ | P@3 | 29.30 | 40.37 | **40.40** | 21.47 | 20.96 | 35.36 | 37.18 | 33.74 |
| $N_{test} = 153025$ | P@5 | 26.12 | 36.92 | **36.96** | 18.61 | 17.72 | 32.04 | 33.85 | 32.64 |
| $d = 135909$ | $T_{train}$ | 422m | 102m | 373h | 162m | 182m | 241m | **8m** | 3134m★ |
| $m = 670091$ | $T_{test}/N_{test}$ | 3.39ms | 66.09ms | 23m | 7.84ms | 5.13ms | **1.72ms** | 0.68ms* | 16.18ms★ |
| | model size | 10G | 6.0G | 3.8G | 3.2G | 3.2G | 1.5G | **0.7G*** | 1.5G★ |

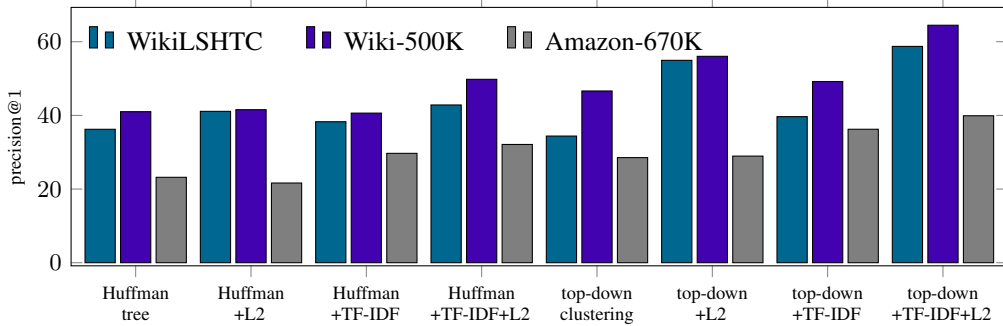

Figure 1: The ablation analysis of different variants of XT on WIKILSHTC, WIKI-500K, and AMAZON-670K.

## Acknowledgements

The work of Kalina Jasinska was supported by the Polish National Science Center under grant no. 2017/25/N/ST6/00747. The work of Krzysztof Dembczyński was supported by the Polish Ministry of Science and Higher Education under grant no. 09/91/DSPB/0651. Computational experiments have been performed in Poznan Supercomputing and Networking Center.

## Footnotes

[1]The derivation is given in Appendix A.

[2]For more detailed introduction to strongly proper composite losses, we refer the reader to (Agarwal, 2014).

[3]Implementation of XT is available at `https://github.com/mwydmuch/extremeText`.

[4] Additional statistics of these datasets are also included in Appendix F. Address of the XMLC repository: http://manikvarma.org/downloads/XC/XMLRepository.html

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
