[Supplementary Material]

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

for any $1 \leq n \leq l$. A similar equation holds for the estimates $\hat{\eta}_j(\boldsymbol{x})$, $\widehat{\mathbf{P}}(z_i \,|\, \boldsymbol{z}^{i-1}, \boldsymbol{x})$, and $\widehat{\mathbf{P}}(\boldsymbol{z}^{n-1} \,|\, \boldsymbol{x})$.

By expressing $\eta_j(\boldsymbol{x})$ and $\hat{\eta}_j(\boldsymbol{x})$ in the aforementioned way we get:

$$
\begin{aligned}
|\eta_j(\boldsymbol{x}) - \hat{\eta}_j(\boldsymbol{x})| &= \left| \prod_{i=0}^{l} \mathbf{P}(z_i \,|\, \boldsymbol{z}^{i-1}, \boldsymbol{x}) - \prod_{i=0}^{l} \widehat{\mathbf{P}}(z_i \,|\, \boldsymbol{z}^{i-1}, \boldsymbol{x}) \right| \\
&= \left| \mathbf{P}(\boldsymbol{z}^{l-1} \,|\, \boldsymbol{x}) \mathbf{P}(z_l \,|\, \boldsymbol{z}^{l-1}, \boldsymbol{x}) - \widehat{\mathbf{P}}(\boldsymbol{z}^{l-1} \,|\, \boldsymbol{x}) \widehat{\mathbf{P}}(z_l \,|\, \boldsymbol{z}^{l-1}, \boldsymbol{x}) \right|
\end{aligned}
$$

By adding and subtracting $\widehat{\mathbf{P}}(z_l \,|\, \boldsymbol{z}^{l-1}, \boldsymbol{x})\mathbf{P}(\boldsymbol{z}^{l-1} \,|\, \boldsymbol{x})$ and using the fact that $|a-b| \leq |a-c|+|b-c|$, and that probability values are in $[0,1]$, we can write:

$$
\begin{aligned}
|\eta_j(\boldsymbol{x}) - \hat{\eta}_j(\boldsymbol{x})| &= \left| \mathbf{P}(\boldsymbol{z}^l \,|\, \boldsymbol{x}) - \widehat{\mathbf{P}}(\boldsymbol{z}^l \,|\, \boldsymbol{x}) \right| \\
&= \left| \mathbf{P}(\boldsymbol{z}^{l-1} \,|\, \boldsymbol{x})\mathbf{P}(z_l \,|\, \boldsymbol{z}^{l-1}, \boldsymbol{x}) - \widehat{\mathbf{P}}(\boldsymbol{z}^{l-1} \,|\, \boldsymbol{x})\widehat{\mathbf{P}}(z_l \,|\, \boldsymbol{z}^{l-1}, \boldsymbol{x}) \right| \\
&= \left| \mathbf{P}(\boldsymbol{z}^{l-1} \,|\, \boldsymbol{x})\mathbf{P}(z_l \,|\, \boldsymbol{z}^{l-1}, \boldsymbol{x}) - \widehat{\mathbf{P}}(\boldsymbol{z}^{l-1} \,|\, \boldsymbol{x})\widehat{\mathbf{P}}(z_l \,|\, \boldsymbol{z}^{l-1}, \boldsymbol{x}) \right. \\
&\qquad + \left. \widehat{\mathbf{P}}(z_l \,|\, \boldsymbol{z}^{l-1}, \boldsymbol{x})\mathbf{P}(\boldsymbol{z}^{l-1} \,|\, \boldsymbol{x}) - \widehat{\mathbf{P}}(z_l \,|\, \boldsymbol{z}^{l-1}, \boldsymbol{x})\mathbf{P}(\boldsymbol{z}^{l-1} \,|\, \boldsymbol{x}) \right| \\
&\leq \left| \mathbf{P}(\boldsymbol{z}^{l-1} \,|\, \boldsymbol{x})\mathbf{P}(z_l \,|\, \boldsymbol{z}^{l-1}, \boldsymbol{x}) - \widehat{\mathbf{P}}(z_l \,|\, \boldsymbol{z}^{l-1}, \boldsymbol{x})\mathbf{P}(\boldsymbol{z}^{l-1} \,|\, \boldsymbol{x}) \right| \\
&\qquad + \left| \widehat{\mathbf{P}}(\boldsymbol{z}^{l-1} \,|\, \boldsymbol{x})\widehat{\mathbf{P}}(z_l \,|\, \boldsymbol{z}^{l-1}, \boldsymbol{x}) - \widehat{\mathbf{P}}(z_l \,|\, \boldsymbol{z}^{l-1}, \boldsymbol{x})\mathbf{P}(\boldsymbol{z}^{l-1} \,|\, \boldsymbol{x}) \right| \\
&= \mathbf{P}(\boldsymbol{z}^{l-1} \,|\, \boldsymbol{x})\left| \mathbf{P}(z_l \,|\, \boldsymbol{z}^{l-1}, \boldsymbol{x}) - \widehat{\mathbf{P}}(z_l \,|\, \boldsymbol{z}^{l-1}, \boldsymbol{x}) \right| \\
&\qquad + \widehat{\mathbf{P}}(z_l \,|\, \boldsymbol{z}^{l-1}, \boldsymbol{x})\left| \widehat{\mathbf{P}}(\boldsymbol{z}^{l-1} \,|\, \boldsymbol{x}) - \mathbf{P}(\boldsymbol{z}^{l-1} \,|\, \boldsymbol{x}) \right| \\
&\leq \mathbf{P}(\boldsymbol{z}^{l-1} \,|\, \boldsymbol{x})\left| \mathbf{P}(z_l \,|\, \boldsymbol{z}^{l-1}, \boldsymbol{x}) - \widehat{\mathbf{P}}(z_l \,|\, \boldsymbol{z}^{l-1}, \boldsymbol{x}) \right| \\
&\qquad + \left| \mathbf{P}(\boldsymbol{z}^{l-1} \,|\, \boldsymbol{x}) - \widehat{\mathbf{P}}(\boldsymbol{z}^{l-1} \,|\, \boldsymbol{x}) \right|
\end{aligned}
$$

We notice that rightmost term corresponds to the absolute value of the difference of probabilities corresponding to one-element shorter code $\boldsymbol{z}^{l-1}$. Therefore we can use recursion and write:

$$
|\eta_j(\boldsymbol{x}) - \hat{\eta}_j(\boldsymbol{x})| \leq \sum_{i=0}^{l} \mathbf{P}(\boldsymbol{z}^{i-1} \,|\, \boldsymbol{x})\left| \mathbf{P}(z_i | \boldsymbol{z}^{i-1}, \boldsymbol{x}) - \widehat{\mathbf{P}}(z_i | \boldsymbol{z}^{i-1}, \boldsymbol{x}) \right|. \tag{6}
$$

Next, we express the above bound in terms of the regret of the strongly proper composite losses. The $(\boldsymbol{x}, z_i)$ pairs are generated i.i.d. according to $\mathbf{P}(\boldsymbol{x}, z_i \,|\, \boldsymbol{z}^{i-1})$. Assume that a node classifier has a form of a real-valued function $f_{\boldsymbol{z}^i}$. Moreover, there exists a strictly increasing (and therefore invertible) link function $\psi : [0,1] \to \mathbb{R}$ such that $f_{\boldsymbol{z}^i}(\boldsymbol{x}) = \psi(\mathbf{P}(z_i \,|\, \boldsymbol{z}^{i-1}, \boldsymbol{x}))$. Recall that the regret of $f_{\boldsymbol{z}^i}$ in terms of a loss function $\ell$ at point $\boldsymbol{x}$ is defined as:

$$
\mathrm{reg}_\ell(f_{\boldsymbol{z}^i} \,|\, \boldsymbol{z}^{i-1}, \boldsymbol{x}) = L_\ell(f_{\boldsymbol{z}^i} \,|\, \boldsymbol{z}^{i-1}, \boldsymbol{x}) - L_\ell^*(\boldsymbol{z}^{i-1}, \boldsymbol{x}),
$$

where $L_\ell(f_{\boldsymbol{z}^i} \,|\, \boldsymbol{z}^{i-1}, \boldsymbol{x})$ is the expected loss at point $\boldsymbol{x}$:

$$
L_\ell(f_{\boldsymbol{z}^i} \,|\, \boldsymbol{z}^{i-1}, \boldsymbol{x}) = \mathbf{P}(z_i \,|\, \boldsymbol{z}^{i-1}, \boldsymbol{x})\ell(1, f_{\boldsymbol{z}^i}(\boldsymbol{x})) + (1 - \mathbf{P}(z_i \,|\, \boldsymbol{z}^{i-1}, \boldsymbol{x}))\ell(-1, f_{\boldsymbol{z}^i}(\boldsymbol{x})),
$$

and $L_\ell^*(\boldsymbol{x})$ is the minimum expected loss at point $\boldsymbol{x}$.

If a node classifier is trained by a learning algorithm that minimizes a strongly proper composite loss, then the bound (6) can be expressed in terms of the regret of this loss function (Agarwal, 2014):

$$
\left| \mathbf{P}(z_i \,|\, \boldsymbol{z}^{i-1}, \boldsymbol{x}) - \psi^{-1}(f_{\boldsymbol{z}^i}) \right| \leq \sqrt{\frac{2}{\lambda}}\sqrt{\mathrm{reg}_\ell(f_{\boldsymbol{z}^i} \,|\, \boldsymbol{z}^{i-1}, \boldsymbol{x})}.
$$

By putting the above inequality into (6), we get

$$
\begin{aligned}
|\eta(\boldsymbol{x}, j) - \hat{\eta}(\boldsymbol{x}, j)| &\leq \sum_{i=0}^{l} \mathbf{P}(\boldsymbol{z}^{i-1} \,|\, \boldsymbol{x})\left| \mathbf{P}(z_i \,|\, \boldsymbol{z}^{i-1}, \boldsymbol{x}) - \widehat{\mathbf{P}}(z_i \,|\, \boldsymbol{z}^{i-1}, \boldsymbol{x}) \right| \\
&= \sum_{i=0}^{l} \mathbf{P}(\boldsymbol{z}^{i-1} \,|\, \boldsymbol{x})\left| \mathbf{P}(z_i \,|\, \boldsymbol{z}^{i-1}, \boldsymbol{x}) - \psi^{-1}(f_{\boldsymbol{z}^i}) \right| \\
&\leq \sum_{i=0}^{l} \mathbf{P}(\boldsymbol{z}^{i-1} \,|\, \boldsymbol{x})\sqrt{\frac{2}{\lambda}}\sqrt{\mathrm{reg}_\ell(f_{\boldsymbol{z}^i} \,|\, \boldsymbol{z}^{i-1}, \boldsymbol{x})}
\end{aligned}
$$

$\square$

The above result shows that the absolute error of estimating the marginal probability of label $j$ can be upperbounded by the regret of the node classifiers on the corresponding path from the root to a leaf. Moreover, for zero-regret (i.e., optimal) node classifiers we obtain an optimal multi-label classifier in terms of estimation of marginal probabilities $\eta_j(\boldsymbol{x})$. This result can be further extended for precision@$k$.

Let us denote a set of the top $k$ labels with respect to the true marginals by $\mathcal{Y}_k$ and a set of the top k labels with respect to predicted marginals by $\hat{\mathcal{Y}}_k$. The conditional regret for precision@$k$ is given then by:

$$\mathrm{reg}_{p@k}(\boldsymbol{h}\,|\,\boldsymbol{x}) = \frac{1}{k}\sum_{i\in\mathcal{Y}_k}\eta_i(\boldsymbol{x}) - \frac{1}{k}\sum_{j\in\hat{\mathcal{Y}}_k}\eta_j(\boldsymbol{x})$$

**Theorem 2.** *For any distribution $\mathbf{P}$ and classifier $\boldsymbol{h}$ delivering estimates $\hat{\eta}_j(\boldsymbol{x})$ of the marginal probabilities of labels, the following holds:*

$$\mathrm{reg}_{p@k}(\boldsymbol{h}\,|\,\boldsymbol{x}) = \frac{1}{k}\sum_{i\in\mathcal{Y}_k}\eta_i(\boldsymbol{x}) - \frac{1}{k}\sum_{j\in\hat{\mathcal{Y}}_k}\eta_j(\boldsymbol{x}) \le 2\max_l|\eta_l(\boldsymbol{x})-\hat{\eta}_l(\boldsymbol{x})|$$

*Proof.* Let us add and subtract the following two terms, $\frac{1}{k}\sum_{i\in\mathcal{Y}_k}\hat{\eta}_i(\boldsymbol{x})$ and $\frac{1}{k}\sum_{j\in\hat{\mathcal{Y}}_k}\hat{\eta}_j(\boldsymbol{x})$, to the regret and reorganize the expression in the following way:

$$\mathrm{reg}_{p@k}(\boldsymbol{h}\,|\,\boldsymbol{x}) = \underbrace{\frac{1}{k}\sum_{i\in\mathcal{Y}_k}\eta_i(\boldsymbol{x}) - \frac{1}{k}\sum_{i\in\mathcal{Y}_k}\hat{\eta}_i(\boldsymbol{x})}_{\le\frac{1}{k}\sum_{i\in\mathcal{Y}_k}|\eta_i(\boldsymbol{x})-\hat{\eta}_i(\boldsymbol{x})|}$$

$$+ \underbrace{\frac{1}{k}\sum_{j\in\hat{\mathcal{Y}}_k}\hat{\eta}_j(\boldsymbol{x}) - \frac{1}{k}\sum_{j\in\hat{\mathcal{Y}}_k}\eta_j(\boldsymbol{x})}_{\le\frac{1}{k}\sum_{j\in\hat{\mathcal{Y}}_k}|\hat{\eta}_j(\boldsymbol{x})-\eta_j(\boldsymbol{x})|}$$

$$+ \underbrace{\frac{1}{k}\sum_{i\in\mathcal{Y}_k}\hat{\eta}_i(\boldsymbol{x}) - \frac{1}{k}\sum_{j\in\hat{\mathcal{Y}}_k}\hat{\eta}_j(\boldsymbol{x})}_{\le 0}$$

Because of the relations given under the braces, we finally get:

$$\mathrm{reg}_{p@k}(\boldsymbol{h}\,|\,\boldsymbol{x}) \le 2\max_l|\eta_l(\boldsymbol{x})-\hat{\eta}_l(\boldsymbol{x})|\ .$$

$\square$

By getting together both theorems we get an upper bound of the precision@$k$ regret expressed in terms of the regret of the node classifiers. Again, for the zero-regret node classifiers, we get optimal solution in terms of precision@$k$.

## B  Hierarchical softmax with the pick-one-label heuristic

**Proposition 2.** *Given conditionally independent labels, a classifier $\boldsymbol{h}$ such that $h_j(\boldsymbol{x}) = \eta'_j(\boldsymbol{x})$ for all $j \in \{1,\ldots,m\}$ has zero regret in terms of the precision@$k$ loss.*

*Proof.* To proof the proposition it suffices to show that for conditionally independent labels the order of labels induced by the marginal probabilities $\eta_j(\boldsymbol{x})$ is the same as the order induced by the values of $\eta'_j(\boldsymbol{x})$ obtained by the pick-one-label heuristic (3):

$$\eta'_j(\boldsymbol{x}) = \mathbf{P}'(y_j = 1\,|\,\boldsymbol{x}) = \sum_{\boldsymbol{y}\in\mathcal{Y}}\frac{y_j}{\sum_{j'=1}^m y_{j'}}\mathbf{P}(\boldsymbol{y}\,|\,\boldsymbol{x}).$$

In other words, for any two labels $i,j \in \{1,\ldots,m\}$, $i \ne j$, $\eta_i(\boldsymbol{x}) \ge \eta_j(\boldsymbol{x}) \Leftrightarrow \eta'_i(\boldsymbol{x}) \ge \eta'_j(\boldsymbol{x})$.

Let $\eta_i(\boldsymbol{x}) \geq \eta_j(\boldsymbol{x})$. The summation over all $\boldsymbol{y}$ in (3) can be written in the following way:

$$\eta_j'(\boldsymbol{x}) = \sum_{\boldsymbol{y} \in \mathcal{Y}} y_j N(\boldsymbol{y}) \mathbf{P}(\boldsymbol{y}|\boldsymbol{x}),$$

where $N(\boldsymbol{y}) = (\sum_{i=1}^m y_i)^{-1}$ is a value that depends only on the number of positive labels in $\boldsymbol{y}$. In this summation we consider four subsets of $\mathcal{Y}$, creating a partition of this set:

$$\mathcal{S}_{i,j}^{u,w} = \{\boldsymbol{y} \in \mathcal{Y} : y_i = u \wedge y_j = w\}, \quad u, w \in \{0, 1\}.$$

The subset $\mathcal{S}_{i,j}^{0,0}$ does not play any role because $y_i = y_j = 0$ and therefore do not contribute to the final sum. Then (3) can be written in the following way for the $i$-th and $j$-th label:

$$\eta_i'(\boldsymbol{x}) = \sum_{\boldsymbol{y}:\mathcal{S}_{i,j}^{1,0}} N(\boldsymbol{y})\mathbf{P}(\boldsymbol{y}|\boldsymbol{x}) + \sum_{\boldsymbol{y} \in \mathcal{S}_{i,j}^{1,1}} N(\boldsymbol{y})\mathbf{P}(\boldsymbol{y}|\boldsymbol{x}) \tag{7}$$

$$\eta_j'(\boldsymbol{x}) = \sum_{\boldsymbol{y}:\mathcal{S}_{i,j}^{0,1}} N(\boldsymbol{y})\mathbf{P}(\boldsymbol{y}|\boldsymbol{x}) + \sum_{\boldsymbol{y} \in \mathcal{S}_{i,j}^{1,1}} N(\boldsymbol{y})\mathbf{P}(\boldsymbol{y}|\boldsymbol{x}) \tag{8}$$

The contribution of elements from $\mathcal{S}_{i,j}^{1,1}$ is equal for both $\eta_i'(\boldsymbol{x})$ and $\eta_j'(\boldsymbol{x})$. It is so because the value of $N(\boldsymbol{y})\mathbf{P}(\boldsymbol{y}|\boldsymbol{x})$ is the same for all $\boldsymbol{y} \in \mathcal{S}_{i,j}^{1,1}$: the conditional joint probabilities $\mathbf{P}(\boldsymbol{y}|\boldsymbol{x})$ are fixed and they are multiplied by the same factors $N(\boldsymbol{y})$.

Consider now the contributions of $\mathcal{S}_{i,j}^{1,0}$ and $\mathcal{S}_{i,j}^{0,1}$ to the relevant sums. By the definition of $\mathcal{Y}$, $\mathcal{S}_{i,j}^{1,0}$, and $\mathcal{S}_{i,j}^{0,1}$, there exists bijection $b_{i,j} : \mathcal{S}_{i,j}^{1,0} \to \mathcal{S}_{i,j}^{0,1}$, such that for each $\boldsymbol{y}' \in \mathcal{S}_{i,j}^{1,0}$ there exists $\boldsymbol{y}'' \in \mathcal{S}_{i,j}^{0,1}$ equal to $\boldsymbol{y}'$ except on the $i$-th and the $j$-th position.

Notice that because of the conditional independence assumption the joint probabilities of elements in $\mathcal{S}_{i,j}^{1,0}$ and $\mathcal{S}_{i,j}^{0,1}$ are related to each other. Let $\boldsymbol{y}'' = b_{i,j}(\boldsymbol{y}')$, where $\boldsymbol{y}' \in \mathcal{S}_{i,j}^{1,0}$ and $\boldsymbol{y}'' \in \mathcal{S}_{i,j}^{0,1}$. The joint probabilities are:

$$\mathbf{P}(\boldsymbol{y}'|\boldsymbol{x}) = \eta_i(\boldsymbol{x})(1 - \eta_j(\boldsymbol{x})) \prod_{l \in \mathcal{L} \setminus \{i,j\}} \eta_l(\boldsymbol{x})^{y_l} (1 - \eta_l(\boldsymbol{x}))^{1-y_l}$$

and

$$\mathbf{P}(\boldsymbol{y}''|\boldsymbol{x}) = (1 - \eta_i(\boldsymbol{x}))\eta_j(\boldsymbol{x}) \prod_{l \in \mathcal{L} \setminus \{i,j\}} \eta_l(\boldsymbol{x})^{y_l} (1 - \eta_l(\boldsymbol{x}))^{1-y_l}.$$

One can easily notice the relation between these probabilities:

$$\mathbf{P}(\boldsymbol{y}'|\boldsymbol{x}) = \eta_i(\boldsymbol{x})(1 - \eta_j(\boldsymbol{x}))q_{i,j} \quad \text{and} \quad \mathbf{P}(\boldsymbol{y}''|\boldsymbol{x}) = (1 - \eta_i(\boldsymbol{x}))\eta_j(\boldsymbol{x})q_{i,j},$$

where $q_{i,j} = \prod_{l \in \mathcal{L} \setminus \{i,j\}} \eta_l(\boldsymbol{x})^{y_l} (1 - \eta_l(\boldsymbol{x}))^{1-y_l} \geq 0$. Consider now the difference of these two probabilities:

$$\begin{aligned} \mathbf{P}(\boldsymbol{y}'|\boldsymbol{x}) - \mathbf{P}(\boldsymbol{y}''|\boldsymbol{x}) &= \eta_i(\boldsymbol{x})(1 - \eta_j(\boldsymbol{x}))q_{i,j} - (1 - \eta_i(\boldsymbol{x}))\eta_j(\boldsymbol{x})q_{i,j} \\ &= q_{i,j}(\eta_i(\boldsymbol{x})(1 - \eta_j(\boldsymbol{x})) - (1 - \eta_i(\boldsymbol{x}))\eta_j(\boldsymbol{x})) \\ &= q_{i,j}(\eta_i(\boldsymbol{x}) - \eta_j(\boldsymbol{x})). \end{aligned}$$

From the above we see that $\eta_i(\boldsymbol{x}) \geq \eta_j(\boldsymbol{x}) \Rightarrow \mathbf{P}(\boldsymbol{y}'|\boldsymbol{x}) \geq \mathbf{P}(\boldsymbol{y}''|\boldsymbol{x})$. Due to the properties of the bijection $b_{i,j}$, the number of positive labels in $\boldsymbol{y}'$ and $\boldsymbol{y}''$ is the same and $N(\boldsymbol{y}') = N(\boldsymbol{y}'')$, therefore we also get $\eta_i(\boldsymbol{x}) \geq \eta_j(\boldsymbol{x}) \Rightarrow \sum_{\boldsymbol{y}:\mathcal{S}_{i,j}^{1,0}} N(\boldsymbol{y})\mathbf{P}(\boldsymbol{y}|\boldsymbol{x}) \geq \sum_{\boldsymbol{y}:\mathcal{S}_{i,j}^{0,1}} N(\boldsymbol{y})\mathbf{P}(\boldsymbol{y}|\boldsymbol{x})$, which by (7) and (8) gives us finally $\eta_i(\boldsymbol{x}) \geq \eta_j(\boldsymbol{x}) \Rightarrow \eta_i'(\boldsymbol{x}) \geq \eta_j'(\boldsymbol{x})$.

The implication in the other side, i.e., $\eta_i(\boldsymbol{x}) \geq \eta_j(\boldsymbol{x}) \Leftarrow \mathbf{P}(\boldsymbol{y}'|\boldsymbol{x}) \geq \mathbf{P}(\boldsymbol{y}''|\boldsymbol{x})$ holds obviously for $q_{i,j} > 0$. For $q_{i,j} = 0$, we can notice, however, that $\mathbf{P}(\boldsymbol{y}'|\boldsymbol{x})$ and $\mathbf{P}(\boldsymbol{y}''|\boldsymbol{x})$ do not contribute to the appropriate sums as they are zero, and therefore we can follow a similar reasoning as above, concluding that $\eta_i(\boldsymbol{x}) \geq \eta_j(\boldsymbol{x}) \Leftarrow \eta_i'(\boldsymbol{x}) \geq \eta_j'(\boldsymbol{x})$.

Thus for conditionally independent labels, the order of labels induced by marginal probabilities $\eta_j(\boldsymbol{x})$ is equal to the order induced by $\eta_j'(\boldsymbol{x})$. As the precision@$k$ is optimized by $k$ labels with the highest marginal probabilities, we have that prediction consisted of $k$ labels with highest $\eta_j'(\boldsymbol{x})$ has zero regret for precision@$k$. $\qquad \square$

## C   Huffman codes for PLTs

In this section we analyze computational cost of PLTs in binary case, i.e. every inner node has two children. We define the cost of a tree as the total expected fraction of instances which are used in the inner nodes and show that for the multi-class case minimization of this cost coincides with minimization of Huffman criteria. However, in the multi-label case, this does no longer hold.

We shall use prefix codes, as it is introduced in Section 4, to identify a path from the root to the leaf. Accordingly, a prefix code $\boldsymbol{z} = (1, z_1, \ldots, z_\ell) \in \mathcal{C}$ determines a path of length $\ell$. We denote the length of code $\boldsymbol{z}$ by $|\boldsymbol{z}|$. The probability to observe a (possibly partial) prefix code is $p_{\boldsymbol{z}^i} = \mathbf{P}(\boldsymbol{z}^i)$ with respect to the data distribution. If we are given a data that consists of $n$ instances, then the expected number of instances that is used to train a node classifier in node $\boldsymbol{z}^i$ is $np_{\boldsymbol{z}^{i-1}}$ (the root node coded by $\boldsymbol{z}^0$ uses all instances for training, i.e., $p_{\boldsymbol{z}^{-1}} = 1$). Thus one can define the expected computational cost of a PLT with a fixed structure (i.e., with a fixed set of prefix codes $\mathcal{C}$), as

$$\sum_{\boldsymbol{z}^i} p_{\boldsymbol{z}^{i-1}} \,, \tag{9}$$

where $\sum_{\boldsymbol{z}^i}$ denotes a sum over all tree nodes, i.e., $\boldsymbol{z}^i$s come from a set $\{\boldsymbol{z}^i : \boldsymbol{z} \in \mathcal{C}, 0 \le i \le |\boldsymbol{z}|\}$. The next proposition defines the relation of (9) and Huffman coding. Huffman coding can be naturally defined in the multi-class case using $\mathbf{P}(y_i = 1)$ as weights.

**Proposition 3.** *Huffman code minimizes the expected computational cost that is given in (9) among binary codes if the data is multi-class.*

*Proof.* Recall that the Huffman code minimizes the following criterion:

$$\sum_{\boldsymbol{z} \in \mathcal{C}} p_{\boldsymbol{z}} |\boldsymbol{z}|.$$

In multi-class setting probability $p_{\boldsymbol{z}^i}$ of each inner node $\boldsymbol{z}^i$ equals to the sum of probabilities of its children, and the objective function given in (9) can be rewritten as

$$\sum_{\boldsymbol{z}^i} p_{\boldsymbol{z}^{i-1}} = 1 + \sum_{\boldsymbol{z} \in \mathcal{C}} p_{\boldsymbol{z}} \sum_{i=1}^{|\boldsymbol{z}|} \deg(\boldsymbol{z}^{i-1}) \,,$$

where $\deg(\boldsymbol{z}^i) = \#\{c : \hat{z}_{i+1} = c, \boldsymbol{z}^i = \hat{\boldsymbol{z}}^i, \hat{\boldsymbol{z}} \in \mathcal{C}\}$ is the number of children of the node $\boldsymbol{z}^i$. In the case of binary codes $\deg(\boldsymbol{z}^i) = 2$ and

$$1 + \sum_{\boldsymbol{z} \in \mathcal{C}} p_{\boldsymbol{z}} \sum_{i=1}^{|\boldsymbol{z}|} \deg(\boldsymbol{z}^{i-1}) = 1 + 2 \sum_{\boldsymbol{z} \in \mathcal{C}} p_{\boldsymbol{z}} |\boldsymbol{z}|$$

which completes the proof.  □

Note that Grave et al. (2017) considered similar notion of computational cost and optimized in a restricted setup, assuming that the tree depth is at most two. Of course, in practice, the $p_{\boldsymbol{z}^i}$ values are not known, but are estimated based on observations from the data distribution. Let us denote an estimate $\hat{p}_{\boldsymbol{z}^i}$ of $p_{\boldsymbol{z}^i}$. An interesting question to address is that to what extent the empirical computational cost $\sum_{\boldsymbol{z}^i} \hat{p}_{\boldsymbol{z}^{i-1}}$ concentrates around its expected values, and on what parameter of the code/tree structure does depend on.

For multi-label case probability $p_{\boldsymbol{z}^i}$ of an inner node cannot be represented by a sum of children probabilities, and this result does not hold.

## D   Pseudocode of PLTs

The pseudocode below presents the training and prediction procedures of PLTs in detail. PLTs can be trained either in the online or batch mode. Algorithm 1 follows the former mode used, for example, in XT, the state-of-the-art variant of PLTs described in Section 7. Note that learning of feature embeddings used in XT is not included in the pseudocode.

In Algorithm 2 we present an inference algorithm, which uses the uniform-cost search for finding the top $k$ labels. The algorithm searches the tree by starting from the root node $z^0$. It uses a priority queue $Q$ to store pairs of the node $z^i$ and its probability estimate $\hat{\eta}_{z^i}$. The algorithm pops the element with the highest probability estimate from the queue. If the element is a leaf we add the corresponding label to the final prediction. Otherwise, the algorithm estimates probabilities of the children nodes and adds them into the queue. Once we found the $k$-th label or we run out of the pairs in the queue, we stop the search procedure.

---

**Algorithm 1** Incremental learning of a PLT:

---

    **Input:**
    $T$: a label tree with $t$ nodes
    $A_{\text{online}}$: an incremental learning algorithm
    $\mathcal{D}_N$: a set of training examples $(\boldsymbol{x}, \boldsymbol{y})$

    **Output:**
    $\mathcal{F}_t = \{f_{z^i}\}^t$: a set of $t$ node (binary) classifiers

1: $\mathcal{F}_t = \emptyset$
2: **for** each node $z^i \in T$ **do**                                      $\triangleright$ Initialization of binary classifiers
3:     $\mathcal{F}_t \leftarrow \mathcal{F}_t \cup$ new classifier $f_{z^i}$
4: **for** each training example $(\boldsymbol{x}, \boldsymbol{y}) \in \mathcal{D}_N$ **do**
5:     **if** $\sum_{j=1}^m y_j = 0$ **then**          $\triangleright$ Select nodes for the positive and negative update
6:         $\mathcal{Z}_{\text{positive}} \leftarrow \emptyset$
7:         $\mathcal{Z}_{\neg\text{positive}} \leftarrow z^0$
8:     **else**
9:         **for** each $y_j = 1, j \in \{1, \ldots, m\}$ **do**
10:             $z \leftarrow$ encode $j$
11:             $\mathcal{Z}_{\text{positive}} \leftarrow \mathcal{Z}_{\text{positive}} \cup \left\{ \bigcup_{i=0}^{l} z^i \right\}$
12:         **for** each $z^i \in \mathcal{Z}_{\text{positive}}$ **do**
13:             $\mathcal{Z}_{\neg\text{positive}} \leftarrow \mathcal{Z}_{\neg\text{positive}} \cup \left\{ \bigcup_{z_i}(z^{i-1}, z_i) \right\} \setminus \mathcal{Z}_{\text{positive}}$
14:     **for** each node $z^i \in \mathcal{Z}_{\text{positive}}$ **do**                    $\triangleright$ Update node classifiers
15:         $f_{z^i} \leftarrow A_{\text{online}}(f_{z^i}, \boldsymbol{x}, 1)$
16:     **for** each node $z^i \in \mathcal{Z}_{\neg\text{positive}}$ **do**
17:         $f_{z^i} \leftarrow A_{\text{online}}(f_{z^i}, \boldsymbol{x}, 0)$
18: **return** $\mathcal{F}_t$.

---

# E    Additional experimental results

## E.1   Comparison of PLTs and HSM on synthetic data

In this section we validate our theoretical results presented in Section 5, which show that HSM is not amenable to model the marginal probabilities in general for multi-label problems. In this case, HSM with pick-one-label heuristic should be outperformed in terms of precision@$k$ by PLTs which are consistent for this performance measure. To validate this claim empirically, we compare the performance of PLTs and HSM on synthetic datasets of three types: multi-label data with independent labels, multi-label data with conditionally dependent labels and multi-class data.

All synthetic models are based on linear models parametrized by a weight vector $\boldsymbol{w}$ of size $d$. The values of the vector are sampled uniformly from a $d$-dimensional sphere of radius 1. Each instance $\boldsymbol{x}$, in turn, is represented as a vector sampled from a $d$-dimensional disc of the same radius.

**Algorithm 2** Top-k prediction with a PLT:

**Input:**
$T$: a label tree with $t$ nodes
$\mathcal{F}_t = \{f_{\boldsymbol{z}^i}\}^t$: a set of $t$ node (binary) classifiers
$\boldsymbol{x}$: a test example
$k$: a size of prediction

**Output:**
$\hat{\boldsymbol{y}}$: a vector with top $k$ labels for the test example

1: $\hat{\boldsymbol{y}} \leftarrow \{0\}^m$
2: $Q \leftarrow \text{PRIORITYQUEUE}()$                         ▷ Initialization of priority queue
3: $\text{add}\big(Q, (1, \boldsymbol{z}^0)\big)$
4: **while** $Q \neq \emptyset$ and $\sum_{j=1}^m \hat{y}_j < k$ **do**            ▷ Check if $k$ labels not found
5:      $\big(\hat{\eta}_{\boldsymbol{z}^i}, \boldsymbol{z}^i\big) \leftarrow \text{pop}(Q)$
6:      **if** $i \in \mathcal{C}$ **then**                      ▷ Check if leaf node reached
7:          $j \leftarrow \text{decode } \boldsymbol{z}^i$
8:          $\hat{y}_j \leftarrow 1$
9:      **else**
10:          **for** each $z_{i+1}$ **do**              ▷ For each node's children
11:              $\boldsymbol{z}^{i+1} \leftarrow (\boldsymbol{z}^i, z_i)$
12:              $\text{add}(Q, (\hat{\eta}_{\boldsymbol{z}^i} \cdot \widehat{\mathbf{P}}_{f_{\boldsymbol{z}^i}}(z_{i+1} \,|\, \boldsymbol{z}^i, \boldsymbol{x}), \boldsymbol{z}^{i+1}))$
13: **return** $\hat{\boldsymbol{y}}$.

**Multi-class distribution.** We associate a weigh vector $\boldsymbol{w}_j$ with each label $j \in \{1, \ldots, m\}$. The model assigns probabilities to labels at point $\boldsymbol{x}$ based on the softmax schema

$$\eta_j(\boldsymbol{x}) = \frac{\exp(c \boldsymbol{w}_j^\top \boldsymbol{x})}{\sum_{j'=1}^m \exp(c \boldsymbol{w}_{j'}^\top \boldsymbol{x})} \tag{10}$$

and draws the positive label according to this probability distribution over labels. Scaling factor $c$ is added to control noise in the model. Higher values of $c$ give less noisy model.

**Multi-label distribution with conditionally independent labels.** The model is similar to the previous one used for the multi-class distribution. The difference lays is normalization as the marginal probabilities do not have to sum up to 1. To get a probability of the $j$-th label, we use the logistic transformation:

$$\eta_j(\boldsymbol{x}) = \frac{\exp(\boldsymbol{w}_j^\top \boldsymbol{x})}{1 + \exp(\boldsymbol{w}_j^\top \boldsymbol{x})}.$$

Then, we assign a label to an instance based on:

$$y_j = [\![r < \eta_j(\boldsymbol{x})]\!],$$

where the random value $r$ is sampled uniformly and independently from range $[0, 1]$ for each instance $\boldsymbol{x}$ and label $j \in \{1, \ldots, m\}$.

**Multi-label distribution with conditionally dependent labels.** To model conditionally dependent labels we use the mixing matrix model based on latent scoring functions generated by $\boldsymbol{W} = (\boldsymbol{w}_1, \ldots, \boldsymbol{w}_m)$. The $m \times m$ mixing matrix $\boldsymbol{M}$ introduces dependencies between noise $\boldsymbol{\epsilon}$, which stands for the source of randomness in the model. The models $\boldsymbol{w}_j$ are sampled from a sphere of radius 1, as in previous cases. The values in the mixing matrix $\boldsymbol{M}$ are sampled uniformly and independently from $[-1, 1]$. The random noise vector $\boldsymbol{\epsilon}$ is sampled from $N(0, 0.25)$. The label vector $\boldsymbol{y}$ is then obtained by element-wise evaluation of the following expression:

$$\boldsymbol{y} = [\![\boldsymbol{M}(\boldsymbol{W}^\top \boldsymbol{x} + \boldsymbol{\epsilon}) > 0]\!]$$

Notice that if $\boldsymbol{M}$ was an identity matrix the model would generate independent labels.

Table 2: Means and standard deviations of 50 runs of each experiment. The p-values on the right indicate the significance of the observed differences. The results of HSM and PLTs on multi-class data were always equal.

| | HSM | | PLT | | | p-values | |
| | mean | stdev | mean | stdev | t-test | sign | Wilcoxon |
|---|---|---|---|---|---|---|---|
| multi-class | 21.90 | 2.74 | 21.90 | 2.74 | | | |
| multi-label independent | 32.57 | 0.34 | 32.58 | 0.33 | 0.4367 | 0.3222 | 0.5980 |
| multi-label dependent | 70.68 | 5.73 | 71.68 | 5.65 | 9.80E-14 | 3.71E-11 | 3.90E-09 |

**Experimental setting.** PLTs and HSM are usually implemented as online learning algorithms, i.e., the node classifiers are updated in an incremental way, example by example. To minimize the impact of the hyperparameter tuning of online algorithms, we have decided to implement batch versions of both algorithms using the LIBLINEAR-based (Fan et al., 2008) logistic regression. The sets of training instances for each node classifier are appropriately constructed by taking the corresponding conditioning of the probabilistic models into account. In the case of HSM with the pick-one-label heuristic, we first transform each multi-label example to $s$ multi-class copies of it, one copy for each its label. Each such copy gets then a weight of $1/s$. Such transformation should be concordant with the theoretical model (3).

In the experiments we used the following parameters of the synthetic models: $d = 3$ (i.e., the number of features), $n = 100000$ instances (split $1 : 1$ for training and test subsets), and $m = 32$ labels or classes. In the case of the multi-class model we report results with the scaling factor $c = 10$. The choice of $c$ does not change the interpretation of the results. To train logistic regression we use a fixed value of the regularization parameter, standing for a very weak regularization. For all experiments we report the results in terms of precision@1.

**Observations.** Table 2 presents the average results of all experiments besides with the standard deviation of obtained values. As expected the performance of PLTs and HSM on the multi-class data are exactly the same. For the other models, we additionally include the p-values of statistical tests run to verify, whether there is a significant difference in performance between PLTs and HSM with the pick-one-label heuristic. In the case of the multi-label data with conditionally independent labels the test shows that there is no evidence to reject the hypothesis that the performance of PLTs and HSM is the same. In the case of multi-label data with conditionally dependent labels, the statistical tests show that PLTs are significantly better than HSM with the pick-one-label heuristic.

### E.2 Comparison of PLTs and HSM on benchmark data

In Table 3 we present similar results, but obtained on the benchmark datasets. We use two models. The first one follows the sparse representation and uses an implementation of PLTs and HSM in VOWPAL WABBIT (Langford et al., 2007). The second one is based on FASTTEXT and produces the dense representation. In all models we use Huffman trees. The results clearly indicate the better performance of PLTs over HSM.

Table 3: Precision@$k$ with $k = \{1, 3, 5\}$ of a simple HSM and a simple PLT implementaions.

| Dataset | VOWPAL WABBIT | | FASTTEXT | | Dataset | VOWPAL WABBIT | | FASTTEXT | |
| | HSM | PLT | HSM | PLT | | HSM | PLT | HSM | PLT |
|---|---|---|---|---|---|---|---|---|---|
| **EUR-Lex** | 56.98 | **74.55** | 66.39 | **73.19** | **AmazonCat-13K** | 86.69 | **91.46** | 90.18 | **92.98** |
| | 46.99 | **60.60** | 54.05 | **57.79** | | 72.00 | **76.00** | 72.53 | **75.75** |
| | 39.09 | **50.05** | 44.73 | **46.98** | | 57.97 | **61.40** | 56.20 | **59.53** |
| **Wiki-30K** | 70.20 | **84.34** | 83.02 | **85.11** | **Delicious-200K** | 41.58 | **45.27** | 42.17 | **46.98** |
| | 60.11 | **72.34** | 69.66 | **73.12** | | 33.24 | **38.95** | 37.94 | **40.99** |
| | 53.17 | **62.72** | 59.50 | **62.67** | | 28.04 | **35.59** | 35.77 | **38.04** |
| **WikiLSHTC-325K** | 36.90 | **41.63** | 41.28 | **41.78** | **Amazon-670K** | 33.64 | **36.85** | 25.04 | **26.18** |
| | 22.30 | **26.78** | 24.68 | **24.96** | | 28.58 | **32.48** | 21.06 | **22.76** |
| | 16.60 | **20.39** | 18.08 | **18.53** | | 25.01 | **29.15** | 18.28 | **20.29** |

### E.3 The ablation analysis on benchmark datasets

Table 4 contains results of the ablation analysis in which we compare different components of the XT algorithm. We analyze the influence of the Huffman tree vs. top-down clustering, the simple averaging of features vectors vs. the TF-IDF-based weighting, and no regularization vs. L2 regularization. For every configuration, we conducted a grid search of hyperparameters from ranges reported in Appendix E.4. The results clearly show that the components need to combined together to obtain the best results. The best combination is usually the one that uses top-down clustering, TF-IDF-based weighting, and L2 regularization. It is worth to notice that top-down clustering alone gets worse results than Huffman trees with TF-IDF-based weighting and L2 regularization.

Table 4: Precision@$k$ scores with $k = \{1, 3, 5\}$ of different variants of PLT in FASTTEXT (EXTREMETEXT)

| Dataset | Metrics | Huff. | Huff. + TF-IDF | Huff. + L2 | Huff. + L2 + TF-IDF | Clus. | Clus. + TF-IDF | Clus. + L2 | Clus. + L2 + TF-IDF |
|---|---|---|---|---|---|---|---|---|---|
| **Eurlex** | P@1 | 63.39 | 71.20 | 62.79 | 74.60 | 68.31 | 75.05 | 65.05 | **77.68** |
| | P@3 | 50.48 | 57.26 | 48.99 | 60.36 | 54.62 | 61.65 | 50.87 | **63.37** |
| | P@5 | 41.19 | 46.93 | 40.16 | 49.72 | 45.23 | 50.99 | 41.71 | **52.85** |
| **AmazonCat-13K** | P@1 | 90.10 | 89.19 | 90.84 | 91.08 | 72.95 | 77.13 | 91.73 | **92.43** |
| | P@3 | 72.67 | 72.81 | 73.10 | 75.27 | 61.66 | 65.76 | 75.07 | **77.65** |
| | P@5 | 57.69 | 58.30 | 57.69 | 60.34 | 48.38 | 52.72 | 59.63 | **62.74** |
| **Wiki-30K** | P@1 | 78.04 | 78.14 | 82.28 | **85.23** | 78.40 | 78.69 | 82.13 | 85.21 |
| | P@3 | 63.31 | 67.47 | 68.75 | **73.52** | 65.62 | 66.40 | 69.22 | 73.18 |
| | P@5 | 52.65 | 56.00 | 58.33 | **63.71** | 55.61 | 56.77 | 58.69 | 63.39 |
| **Delicious-200K** | P@1 | 45.71 | 47.24 | 46.48 | **47.85** | 44.58 | 45.13 | 46.55 | 47.31 |
| | P@3 | 40.69 | 40.88 | 41.38 | **42.08** | 40.42 | 40.63 | 41.08 | 41.63 |
| | P@5 | 38.02 | 37.74 | 38.78 | **39.13** | 38.15 | 38.23 | 38.25 | 38.88 |
| **WikiLSHTC-325K** | P@1 | 36.23 | 38.28 | 41.10 | 42.83 | 34.39 | 39.66 | 54.95 | **58.73** |
| | P@3 | 20.60 | 22.43 | 25.62 | 26.33 | 21.17 | 24.79 | 36.42 | **39.24** |
| | P@5 | 14.70 | 16.33 | 19.10 | 19.45 | 15.09 | 18.28 | 27.25 | **29.26** |
| **Wiki-500K** | P@1 | 41.01 | 40.62 | 41.55 | 49.80 | 46.63 | 49.20 | 56.04 | **64.48** |
| | P@3 | 24.79 | 25.68 | 25.94 | 32.39 | 31.59 | 34.52 | 38.87 | **45.84** |
| | P@5 | 18.64 | 19.74 | 19.79 | 24.62 | 24.30 | 26.83 | 30.46 | **35.46** |
| **Amazon-670K** | P@1 | 23.19 | 29.70 | 21.64 | 32.11 | 28.54 | 36.24 | 28.95 | **39.90** |
| | P@3 | 19.80 | 25.84 | 18.11 | 27.77 | 25.52 | 32.15 | 25.74 | **35.36** |
| | P@5 | 17.48 | 22.96 | 15.83 | 24.64 | 23.30 | 29.19 | 23.22 | **32.04** |

### E.4 Tuning of hyperparameters

The PLT has only one global hyperparameter which is the degree of the tree denoted by $b$. The other hyperparameters are associated with the node classifiers. The HSM and PLT in Vowpal Wabbit was tuned with the stochastic gradient descent with a step size $\eta_t$ calculated separately for each node according to $\eta_t = \eta \times (1/t)^p$ where $t$ is number of node updates and $\eta$ and $p$ are hyperparameters. In FASTTEXT-based methods, HSM, LEARNED TREE and XT, $\eta_t$ decreased linearly during training from $\eta$ to 0.0. In the XT the optimization methods has been extended by L2 regularization, so it has one additional parameter. Balanced k-means clustering used to build a tree in XT has also a stopping parameter $\epsilon$ set by default to 0.001.

Table 5: The hyperparameters of the HSM and PLT methods and their ranges used in hyperparameter optimization. Notation: $b$ – tree arity, $\eta$ – learning rate

| Hyperparameter | Range |
|---|---|
| $b$ | $\{2, \ldots, 32\}$ |
| $\eta$ | $[0.0001 - 1.0]$ |
| number of epochs | $\{20, 30, 40\}$ |

## F  Information about the benchmark datasets

Table 8 contains the basic statistics of the benchmark datasets used in the experiments taken from the Extreme Classification Repository: `http://manikvarma.org/downloads/XC/XMLRepository.html`.

Table 6: The hyperparameters of the FASTTEXT and LEARNED TREE methods and their ranges used in hyperparameter optimization. Notation: $b$ – tree arity, $\eta$ – learning rate

| Hyperparameter | Range |
|---|---|
| $b$ | $\{2, \ldots, 32\}$ |
| $\eta$ | $[0.0001 - 1.0]$ |
| number of epochs | $\{20, 30, 40\}$ |
| dim | $\{500\}$ |

Table 7: The hyperparameters of the XT method and their ranges used in hyperparameter optimization. Notation: $b$ – tree arity (number of centroids used in k-means clustering), $\epsilon$ – stoping condition of k-means clustering, $\eta$ – learning rate

| Hyperparameter | Range |
|---|---|
| $b$ | $\{2\}$ |
| $\epsilon$ | $\{0.0001\}$ |
| $\eta$ | $\{0.05, 0.1, 0.2, 0.3, 0.4, 0.5\}$ |
| L2 regularization | $\{0.001, 0.002, 0.003\}$ |
| number of epochs | $\{20, 30, 40\}$ |
| dim | $\{500\}$ |
| max leaves | $\{100\}$ |

Table 8: Statistics of the benchmark datasets.

| Dataset | Number of features | Number of labels | Number of train points | Number of test points | Avg. points per label | Avg. labels per point |
|---|---|---|---|---|---|---|
| EURLex-4K | 5000 | 3993 | 15539 | 3809 | 25.73 | 5.31 |
| AmazonCat-13K | 203882 | 13330 | 1186239 | 306782 | 448.57 | 5.04 |
| Wiki10-31K | 101938 | 30938 | 14146 | 6616 | 8.52 | 18.64 |
| Delicious-200K | 782585 | 205443 | 196606 | 100095 | 72.29 | 75.54 |
| WikiLSHTC-325K | 1617899 | 325056 | 1778351 | 587084 | 17.46 | 3.19 |
| Wikipedia-500K | 2381304 | 501070 | 1813391 | 783743 | 24.75 | 4.77 |
| Amazon-670K | 135909 | 670091 | 490449 | 153025 | 3.99 | 5.45 |