[Reviews · NeurIPS 2018]

Reviewer 1



The contributions of this paper are: - Provide counter-examples to prove the hierarchical softmax with pick-one-label heuristic is not consistent unless the strong independent assumption is made - The probabilistic label tree is a generalization of the hierarchical softmax when Precision@k is used in the loss. Based on this reduction, a no-regret bound can be proved. - The authors provide a new implementation of PLTs and demonstrate its strong performance. Pros. - The paper is nicely written, providing nice motivation and clear explanations. - The paper provides a comprehensive study and makes contributions in both theory and practice perspectives. - Extreme classification is an important problem, and the paper provides a better understanding of the softmax method. Cons. - Although I found the results in the paper are interesting, the key contributions of this paper are mainly based on [17]. The extension of the no-regret proof of PLT with Precision@k metric is relatively straightforward. - The discussion is limited to only when Precision@k metric is used. Other comments: - Comparing the experimental results demonstrate in the paper with [17], is there a typo of FastXML on Amazon dataset in P@1? The results show in previous work is around 36.65, but this paper shows 54.1. - XT is significantly worse than DISMEC on most datasets except Delicious-200K. Is there a specific situation that XT is expected to perform better?

Reviewer 2



Summary: This work investigates Probabilistic Label Trees (PLTs) in solving extreme multi-label classification (XMLC). The theoretical analysis shows PLT is a no-regret algorithm for precision@k, and the algorithmic improvement combines PLT and fastText to efficiently handle extreme multi-label text classification problems, with a clustering-based tree structure building strategy. This paper is comphrensive and well-written, including extensive experiments. The theory part formally shows PLT outputing k labels with highest marginal probabilities is consistent with precision@k, given zero-regret node classifiers. The authors also provide some negative result on heuristic strategies, one is that pick-one-label heuristic is suboptimal in terms of precision@k, and another is that building Huffman trees for PLT does not minimize computational cost. I think the consistency result is a bit straightforward since the zero-error assumption on marginal probabilities is strong enough to provide many nice properties. However, as the authors said in Line 231-233, if this assumption holds, there is no need to care about the tree structure in PLT, which is surely an important issue in practice. Also, it is easy to see that PLT is preferred than vanilla HSM in XMLC. Personally speaking, I would like to see the authors put more theory parts into the supplementary and show more experiments on the tree structure decision in PLT. For example: - The performance difference between original PLT paper vs. Huffman tree vs. top-down hierarchical clustering used in this paper. - The improvement by applying fastText into PLT. Does fastText help both in time cost and performance? ============================== Post-author feedback comments: I have read the author feedback and found out that their theoretical results are based on the maximum estimation error of marginal probability on nodes. Although this measure may not be available in practice, the theoretical contribution is more than what I expected before. Therefore I increased my score.

Reviewer 3



1. Quality: The proofs clearly demonstrate that pick-one HSM methods are not ideal for XMLC. However, the primary reasoning for using pick-one HSM is for speed. This paper presents a theoretical argument for using PLT for XMLC which performs better than FastXML with about a 3x speedup at test time. However, the results aren’t very significant over the Parabel model or XML-CNN. The experimental results section is incomplete. There should be an explanation of the differences between the datasets (e.g. distribution of positive labels), and an explanation of why XT works better on some datasets. 2. Clarity The formal background was insufficient. Background work wasn’t explained well (especially FastXML, and PLT). F1, Hamming, P@k weren’t explained why they are important for XMLC. P@k is just one measure for XMLC, but many more metrics are important. Minimizing the regret for P@k shouldn’t be the only metric for a good model. It’s somewhat hard to understand the technical contributions of this work. The contributions aren’t clearly outlined, and it is not very informative. 3. Originality The originality of this work is essentially just from proving PLTs are better than pick-one HSMs. The theorems and proofs are new for XMLC, which is important since pick-one HSMs are frequently used in practice. However, there is a very little technical contribution in terms of the method. It is a small improvement over the FastXML model, albeit the results are better than FastXML. In addition, the method is not explained very well - it is limited to one paragraph. 4. Significance XMLC is a very important task, and the proofs in this paper seem to be significant. However, I’m not sure the experimental results and presentation of the work are significant enough.